# Micro-Structure and Mechanical Properties of 2A97 Al-Li Alloy Cast by Low-Frequency Electromagnetic Casting

**Fuyue Wang \*, Xiangjie Wang and Jianzhong Cui**

Key Lab of Electromagnetic Processing of Materials, Ministry of Education, Northeastern University, 314 Mailbox, Shenyang 110819, China
* Correspondence: NEUWFY@sina.com; Tel.: +86-138-4054-2390

**Abstract:** In this paper, the micro-structure and solid solubility of 2A97 aluminum-lithium alloy ingots prepared by conventional direct chill (DC) casting and low frequency electromagnetic casting (LFEC) as well as the density of precipitates and tensile properties of 2A97 alloy after T6 treatment were investigated. The results show that the low-frequency electromagnetic field during casting had a remarkable effect on the refinement of micro-structure and solid solubility of alloying elements within the grains, which is very helpful to obtain a large number of fine, uniformly-distributed $T_1$ phases. The mechanical properties of 2A97 alloy prepared by low frequency electromagnetic casting after T6 treatment were improved by obtaining a larger number $T_1$ phases.

**Keywords:** LFEC (low-frequency electromagnetic casting); 2A97; Al-Li alloy; micro-structure; solid solubility; tensile properties

## 1. Introduction

Aluminum-lithium alloys are very important to the aerospace and spacecraft industries due to their high specific strength, high specific stiffness, high elastic modulus [1,2]. Further, the third generation Al-Li alloys are being increasingly substituted for conventional 2xxx and 7xxx series aluminum alloys [1]. The 2A97 alloy is the first Al-Li alloy enrolled in China and it has become one of the most potential alloys among the third generation Al-Li alloys since it possesses high stiffness, low density, corrosion resistance, and superior damage tolerance.

The preparation of a high metallurgical quality Al-Li alloy ingot is the foundation and precondition of obtaining high performance. Because the molten Al-Li alloys are easily oxidized and prone to the absorption of hydrogen [3,4], the melting and casting of high-metallurgical quality Al-Li is extremely difficult. The most commonly used technology in the production of aluminium-alloy semi-finished products is direct chill (DC) casting, but this process is prone to casting defects in Al-Li alloy ingots including oxidation slags, coarse grains, and alloying element segregation which greatly deteriorates the characteristic properties of the alloys [5–7].

A new casting technique, called the low frequency electromagnetic casting (LFEC) process, was developed by Cui and co-workers [8]. LFEC was implemented with frequencies lower than those in the casting, refining and electromagnetic process (CREM) put forward by Vives [9]. In the LFEC process, a low frequency electromagnetic field was used to control the flow and temperature fields on the sump for obtaining a reasonable solidification condition. With the development of this process, it continues to gain popularity in the preparation of aluminum alloys. The numerous advantages of this process, which include refined micro-structures [10,11], improved surface quality of ingots, enhanced solid solubility of alloying elements within grain interiors [12], and improved macro-segregation and eliminating cracks [13], could improve the metallurgical quality of Al-Li alloys effectively.

Until now, there have been few reports on the Al-Li alloys prepared by DC casting. Thus, in this work, the 2A97 Al-Li alloy ingots were prepared by conventional DC casting and LFEC. In order to investigate the effects of LFEC on the micro-structure, solid solubility, and mechanical properties of 2A97 Al-Li alloy.

## 2. Experimental

### 2.1. Material Preparation

The high purity of Al ingot (99.99%), Mg ingot (99.9%), Li ingot (99.9%), and Al-20Cu, Al-10Mn, Al-5Zr, A1-5Ti-B, A1-3Be master alloys were selected as raw materials. The nominal composition of 2A97 alloy from GB/T3190-2008 is shown in Table 1. The analysed chemical composition of the as-cast alloy ingots were determined by an inductively-coupled plasma atomic emission spectroscope (ICP-AES), also listed in Table 1.

**Table 1.** Nominal and analysed chemical composition of 2A97 alloy (wt %).

| Element | Cu | Li | Mg | Mn | Zr | Ti | Be | Al |
|---------|------|------|------|------|------|------|------|------|
| Nominal | 2.0–3.2 | 0.8–2.3 | 0.25–0.8 | 0.20–0.60 | 0.08–0.20 | 0.001–0.10 | - | Bal. |
| Analysed | 3.05–3.26 | 1.35–1.40 | 0.38–0.45 | 0.38–0.42 | 0.10–0.12 | 0.05 | 0.005 | Bal. |

The material preparation was divided into two processes: melting and casting. The melting process was carried out in a 15-kW resistance furnace with a stainless crucible and protected by an argon atmosphere. Firstly, the Al ingot was put in the resistance furnace at the temperature of 760 °C in order to completely melt the Al ingot. Then, the Al-20Cu, Al-10Mn, Al-5Zr, A1-5Ti-B, and A1-3Be master alloys were added to the melt. The melt temperature decreased to about 720 °C after the added master alloys were completely melted, then the Mg ingot was pressed into the melt by a TC4 titanium alloy cover when the melt temperature was 720 °C before the melt was degassed by $C_2Cl_6$ and the dross was cleared off. The Li ingot was pressed into the melt by the TC4 titanium alloy cover 'when the melt temperature fell to 700 °C, and then molten salt (60% LiCl and 40% LiF) was sprinkled over the surface of the melt as a covering agent to minimize the loss of Li. When the melt temperature stabilized at 695 °C, the melting process was finished and the LFEC casting process was ready to start. The melt was transferred from the furnace through the calcium silicate hot top into the casting crystallizer mold by controlling the flow control stick at 695 °C before being cast into ingots with a diameter of 125 mm at a casting velocity of 110 mm/min and a cooling water supply of 40 L/min. At the same time, a low-frequency electromagnetic field control unit had been started, the frequency was kept at 10 Hz, and the current intensity was varied from 0 A, 50 A, 100 A, to 150 A. The electromagnetic field was applied by a 100 turns copper coil that surrounded the aluminum alloy crystallizer mold. A graphite ring with an inside diameter of 128 mm and height of 20 mm was embedded in the mold, and a calcium silicate hot top was installed on top of the graphite ring. A schematic diagram of the low-frequency electromagnetic casting process is shown in Figure 1.

The 2A97 alloy ingots were homogenized at 500 °C for 24 h. The surface layer of about 5 mm of the ingot was removed and the ingot was subsequently extruded at 380 °C into rods with a diameter of 16 mm before the solution was treated at 500 °C for 90 min in a salt bath furnace and quenched into cold water. The solution was finally aged at 160 °C for 24 h (T6 treatment).

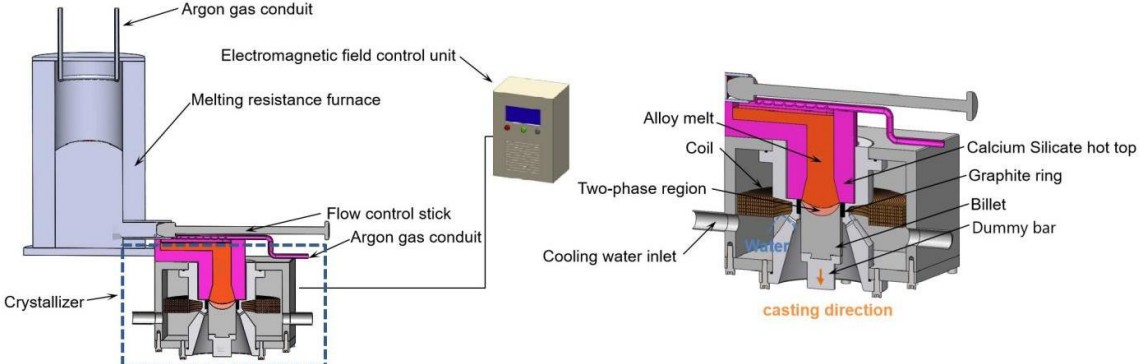

**Figure 1.** Schematic diagram of low-frequency electromagnetic casting process.

*2.2. Characterizations and Testing*

In this research, the effects of LFEC on the micro-structure, solid solubility, and mechanical properties of the 2A97 Al-Li alloy were investigated using an optical microscope (OM), electron probe micro analyzer (EPMA), scanning acoustic microscope (SAM), differential scanning calorimeter (DSC), X-ray diffractometer (XRD), transmission electron microscope (TEM), and tensile testing machine. The as-cast specimen taken from a one-half radius region of cross section of ingots was anodized with 10 vol% $HBF_4$ solution with a voltage of 20 V and a current of 0.5–1.5 mA for about 20 s and then the micro-structure was observed by a Leica (Wetzlar, Germany) DMI5000M optical microscope with cross-polarized light. Specimens for SAM testing were cut from the ingots with 20 mm thickness. The casting defects in the different layers of SAM specimens were scanned by a KSI (Herborn, Germany) V-700E scanning acoustic microscope with a frequency of ultrasonic of 50 MHz. The number and areal fraction of casting defects (bright white spots) in the 4 mm, 8 mm, 12 mm, and 16 mm layer were measured by Image-Pro Plus software (6.0, Media Cybernetics, Rockville, MD, USA). The element mappings and the casting defects of the as-cast specimens were determined by JXA-8530 (JEOL, Tokyo, Japan) electron micro-probe analysis. The micro-structure of central part of the extruded rods after aging were observed by OM and TEM. TEM observation was conducted on a Tecnai $G^2$20 transmission electron micro-scope operating (FEI, Waltham, MA, USA) at 200 kV. Specimens for TEM testing were mechanically thinned to 80–100 μm and then finally thinned via a twin-jet polishing in a solution of 30 vol% $HNO_3$ and 70 vol% methanol at −20 °C under a voltage of 15 V. The length and number density of $T_1$ phase were measured from five random TEM micro-graphs by Image-Pro Plus software. The differential scanning calorimetry measurements were conducted using a Netzsch 404F3 (Netzsch, Germany) differential scanning calorimeter with a constant heating rate of 10 °C/min from 25 °C to 550 °C. The X-ray diffraction measurements were also applied for the phase analysis. An Pw3040/60X X-ray diffractometer (PANalytical B.V., The Netherlands) was operated in asymmetric 2θ scan mode with Cu Kα radiation (wave length λ = 0.15406 nm). The length step was 0.02°, the scanning speed was 1°/min, and the scanning ranged from 20° to 120°. The alloy rods after aging were machined according to ASTM B557M standard. Tensile test was carried out using a AG-Xplus electron testing machine (Shimadzu, Japan) at a strain rate of 2.0 mm/min. The loading axis of the specimen was aligned with the direction of final extrusion. This tensile test was repeated using four specimens for each condition, and then average values of the yield strength, ultimate tensile strength, and elongation were obtained.

## 3. Results

*3.1. Effect of Electromagnetic Field on Micro-Structure of As-Cast Alloys*

The micro-structure of a one-half radius region of cross section of ingots prepared in different electromagnetic conditions is shown in Figure 2. The average grain size of these ingots is shown in Figure 3. In the case of conventional DC casting, grains are found to be mainly characterized

by coarse rose-shape dendrites with an average grain size of 303.2 μm, which exhibit an obvious non-uniform grain size, as shown in Figure 2a. However, the average grain size decreases remarkably with the increased current intensity. The average grain sizes decreases from 203.2 μm to 119.6 μm, respectively corresponding to 50 A and 150 A. When the current intensity was increased to 150 A, the micro-structure containing fine dendritic grains were observed, as shown in Figure 2d.

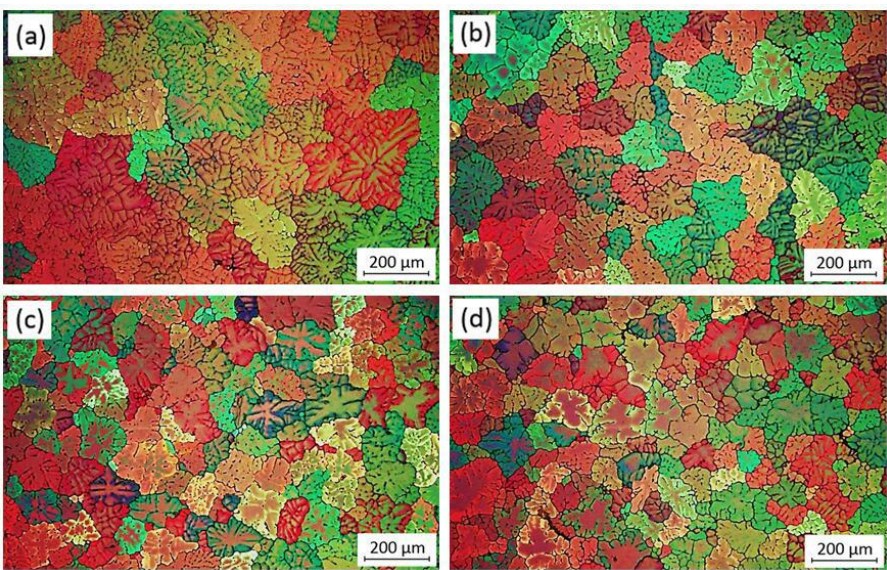

**Figure 2.** Micro-structure of one-half radius region of cross section of ingots prepared in different electromagnetic conditions (**a**) Conventional DC casting; (**b**) 10 Hz/50 A; (**c**) 10 Hz/100 A; (**d**) 0 Hz/150 A.

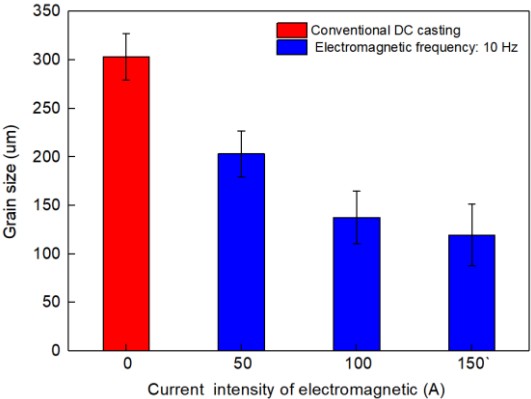

**Figure 3.** Average grain sizes of one-half radius region of cross section of ingots prepared in different electromagnetic conditions.

*3.2. Effect of Electromagnetic Field on Defects of As-Cast Alloys*

The casting defects in the different layers of SAM specimens were scanned by scanning acoustic microscope (SAM). The bright white spots in Figure 4 could reflect the number and distribution of defects. The white annular area in the edge of the ingot was the defect zone, which would have been mechanically turned off after homogenization annealing. Except for the annular defect zone in the marginal area of ingot, the core zone was researched as the defect target area. It is evident from Figure 4 that the areal fraction of casting defects and a number of casting defects of the ingot prepared by conventional DC casting is much greater as compared to that by LFEC with 10 Hz/150 A. The defects were concentrated in the one-half radius region of the cross section of the ingot cast by conventional casting, while the defects are randomly distributed in the cross section of the ingot cast by LFEC with

10 Hz/150 A. The number and areal fraction of casting defects were measured by Image-Pro Plus software, and the results are shown in Table 2.

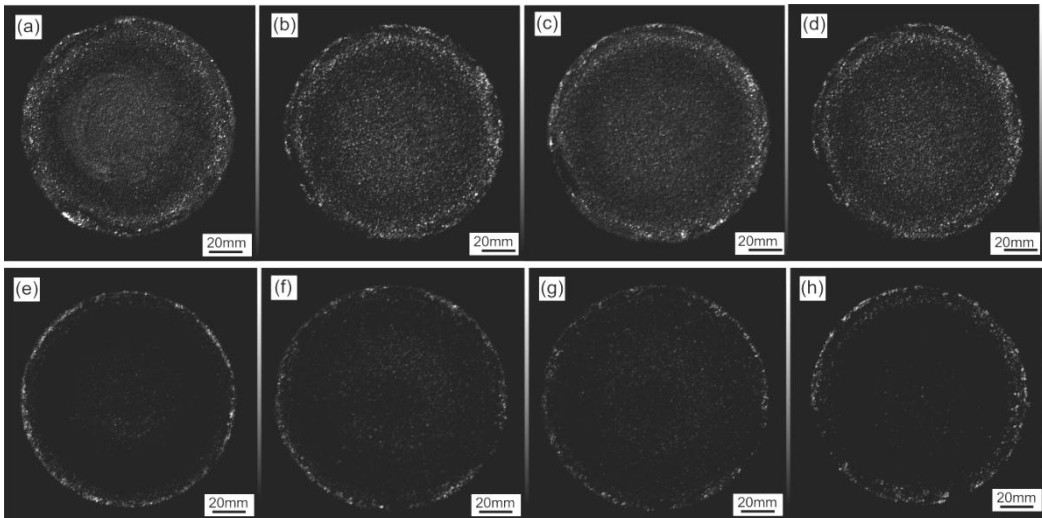

**Figure 4.** Scanning acoustic microscope (SAM) images in the cross section of 2A97 ingot prepared by conventional DC cast and LFEC with 10 Hz/150 A (**a**) DC-4 mm layer; (**b**) DC-8 mm layer; (**c**) DC-12 mm layer; (**d**) DC-16 mm layer; (**e**) LFEC-4 mm layer; (**f**) LFEC-8 mm layer; (**g**) LFEC-12 mm layer; (**h**) LFEC-16 mm layer.

**Table 2.** The statistical results of the casting defects in the SAM images of 2A97 as-cast ingots.

| Casting Condition | Areal Fraction of Defects | Number of Defects |
| --- | --- | --- |
| DC casting | 0.45 ± 0.29% | 149.7 ± 45.1 |
| 10 Hz/150 A | 0.10 ± 0.04% | 33.0 ± 12.6 |

Further, the element maps of the as-cast alloys were determined by EPMA. Typical defects such as the looseness of the structure and oxide inclusion in the ingot prepared by conventional DC are shown in Figure 5. Few defects are shown in the ingots prepared by LFEC with 150 A/10 Hz in Figure 6. Figure 5a shows the morphologies of the rose-shaped eutectic-structure inhomogeneously distributed along grain boundaries, however it became a continuous thin fishing net eutectic structure as shown in Figure 6a, in which Cu and Mg tend to be concentrated together and enriched in the fine eutectic phases at grain boundaries. Mn tends to uniformly distributed within grains.

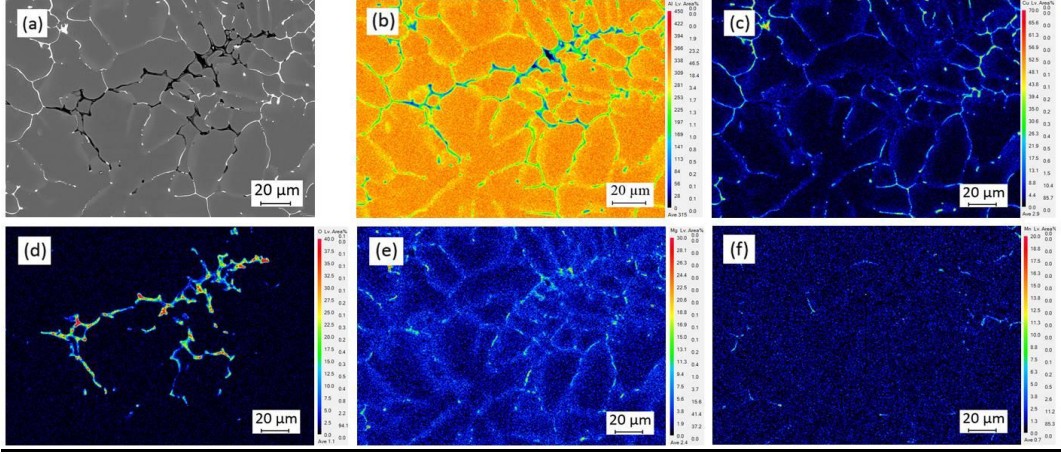

**Figure 5.** Elements map-scanning of 1/2 radius region of as-cast alloy cast by conventional DC casting (**a**) SEM micrograph of as-cast alloy, (**b**) Al; (**c**) Cu; (**d**) O; (**e**) Mg; (**f**) Mn.

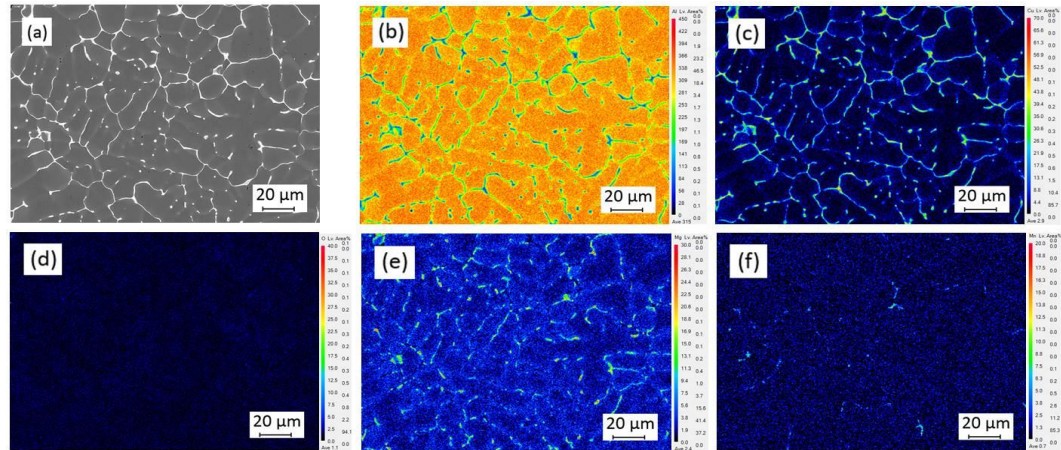

**Figure 6.** Elements map-scanning of 1/2 radius region of as-cast alloy cast by LFEC with 10 Hz/150 A
(**a**) SEM micrograph of as-cast al loy; (**b**) Al; (**c**) Cu; (**d**) O; (**e**) Mg; (**f**) Mn.

### 3.3. Effect of Low-Frequency Electromagnetic Field on Solid Solubility of Alloying Element within Grains of As-Cast Alloys

From the viewpoint of thermodynamics, the dissolution of low melting non-equilibrium eutectic phase in the grain boundary accompanied by the change of the enthalpy [14], this process can be exhibited and analyzed by differential scanning calorimetry (DSC) measurement. The DSC curves of as-cast alloy prepared under different electromagnetic field condition are shown in Figure 7.

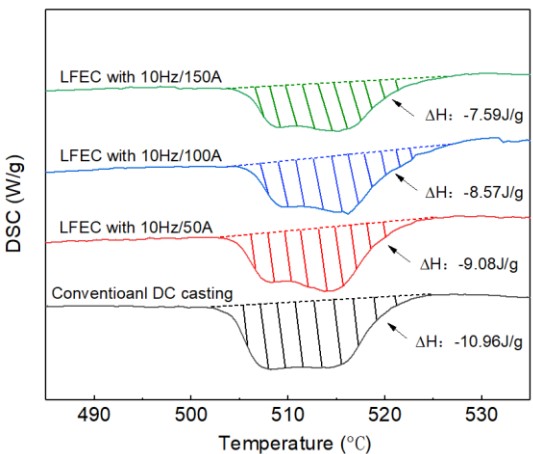

**Figure 7.** DSC curves of 2A97 as-cast alloy cast by the different electromagnetic conditions.

It could be seen from Figure 7 that the position and profile of the endothermic peaks almost not change, which indicates that the composition of the non-equilibrium eutectic phase does not change with the application of the low-frequency electromagnetic field. But area of the endothermic peaks change obviously. By integrating the area of the endothermic peaks in the different condition, the enthalpy (ΔH) of dissolved the non-equilibrium eutectic phase can be calculated quantitatively, and then the amount of the eutectic phase can be expressed. The maximum enthalpy is measured to be −10.96 J/g in the case of conventional DC casting. By comparison, the minimum enthalpy is obtained in the case of LFEC with 10 Hz/150 A for −7.59 J/g. The enthalpy of the dissolved non-equilibrium eutectic phase decreases remarkably with an increase in electromagnetic field current intensity. This suggests that a decrease in the amount of non-equilibrium eutectic phase in the grain boundary and an increase in the amount of the alloying elements within grains are associated with an increasing current intensity of the electromagnetic field.

From the viewpoint of crystallography, based on the position of the solution atom in the crystal lattice, the solid solution can be classified into two types: substitution solid solution and interstitial solid solution. Regardless of the type of solid solution produced in alloy, it will necessarily lead to lattice distortion, requiring extra lattice distortion energy. The extent of lattice distortion can be estimated by evaluating the change in lattice parameters [15–17]—for 2A97 alloy, the main alloying element is Cu since the lattice parameter and atomic size of Cu is smaller than that of Al. Upon the formation of the substitution solid solution, there is a decrease in lattice parameters of $\alpha$(Al). The greater the amount of solid solution of Cu atoms, the greater the magnitude of lattice distortion will be caused, and the larger the change in the lattice parameter of $\alpha$(Al) will be.

Figure 8 illustrates the shift of the Bragg angles of $\alpha$(Al) in the XRD results of 2A97 alloy under the influence of different magnetic field conditions. On (111), (200), (220) and (311) crystal faces of $\alpha$(Al), the value of $2\theta$ corresponding to each Bragg angle creates a larger angle with the increase in current intensity. The Bragg's law of crystal diffraction is expressed as shown in Equation (1).

$$2d\sin\theta = \lambda \tag{1}$$

where $d$ is the interplanar distance; $\theta$ is the angle between the incident X-ray and corresponding crystal plane, and $\lambda$ is the wave length of Cu K$\alpha$ radiation. The relationship between the interplanar distance of the face-centered cube and lattice parameter is given by Equation (2).

$$d = \frac{a}{\sqrt{h^2 + k^2 + l^2}} \tag{2}$$

where $a$ is the lattice parameter, and ($h$ $k$ $l$) are the values for crystal plane index corresponding to the crystal face.

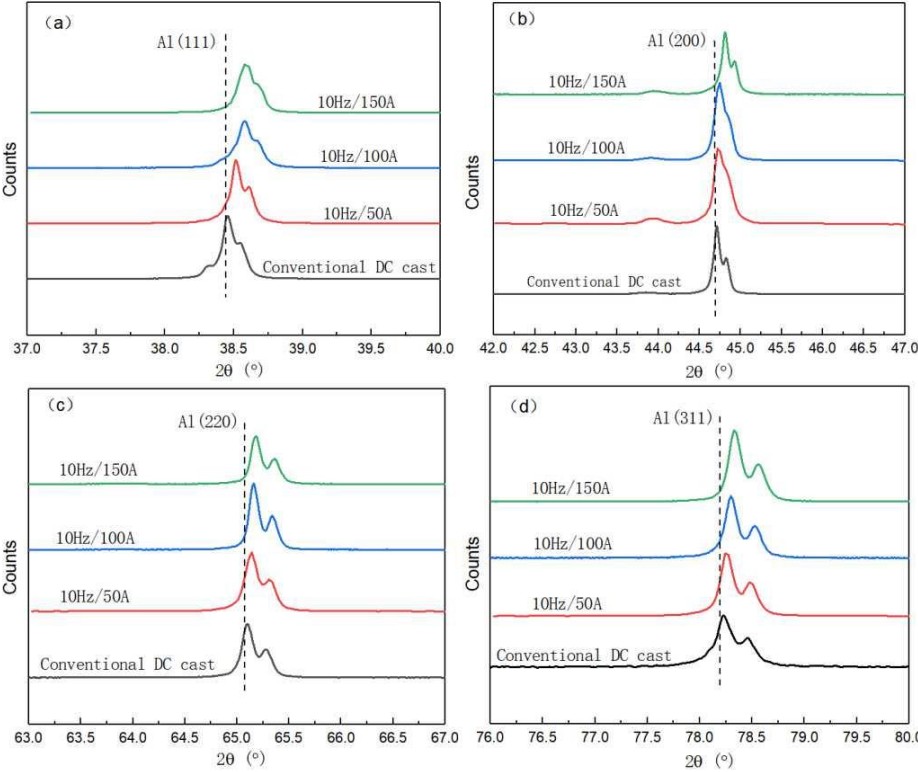

**Figure 8.** Diffraction peaks of $\alpha$(Al) at different crystal faces of 2A97 alloy DC casting and LFEC (**a**) (111); (**b**) (200); (**c**) (220); (**d**) (311).

Table 3 enlists the variation in interplanar distance and lattice parameter based on the Bragg angle within four different crystal faces. Δa is the variation of lattice parameter, and it reflects the extent of lattice distortion. It is noteworthy that on the four different crystal faces of α(Al)—(111), (200), (220) and (311)—the lattice parameter decreases with an increase in the current intensity of the magnetic field.

**Table 3.** Lattice parameters of α(Al) of 2A97 alloy DC casting at different electromagnetic field conditions.

| Miller Indices | Magnetic Field | 2θ/° | a/nm | Δa/nm |
|---|---|---|---|---|
| (111) | DC casting | 38.4872 | 4.0481 | −0.0019 |
| | 10 Hz/50 A | 38.5062 | 4.0462 | −0.0038 |
| | 10 Hz/100 A | 38.5782 | 4.0389 | −0.0110 |
| | 10 Hz/150 A | 38.5777 | 4.0389 | −0.0110 |
| (200) | DC casting | 44.7142 | 4.0502 | −0.0002 |
| | 10 Hz/50 A | 44.7217 | 4.0495 | −0.0005 |
| | 10 Hz/100 A | 44.7487 | 4.0472 | −0.0028 |
| | 10 Hz/150 A | 44.8102 | 4.0419 | −0.0080 |
| (220) | DC casting | 65.1062 | 4.0491 | −0.0009 |
| | 10 Hz/50 A | 65.1427 | 4.0471 | −0.0029 |
| | 10 Hz/100 A | 65.1622 | 4.0460 | −0.0040 |
| | 10 Hz/150 A | 65.1862 | 4.0447 | −0.0053 |
| (311) | DC casting | 78.2342 | 4.0494 | −0.0006 |
| | 10 Hz/50 A | 78.2437 | 4.0490 | −0.0010 |
| | 10 Hz/100 A | 78.2982 | 4.0466 | −0.0038 |
| | 10 Hz/150 A | 78.3302 | 4.0452 | −0.0048 |

In Table 4, the average values of the lattice parameters of α(Al) and the variation in lattice parameters of the 2A97 alloy under different electromagnetic fields are given. It is evident from the XRD results that the amount of solid solubility of alloying elements is greatly enhanced with an increase in electromagnetic field current intensity. Further, this conclusion is consistent with the previous DSC conclusion in Figure 7.

**Table 4.** Effects of electromagnetic field on Lattice parameters of α(Al) of 2A97 alloy.

| Magnetic Field | ā/nm | $\overline{\Delta a}$/nm |
|---|---|---|
| DC casting | 4.0492 | −0.0009 |
| 10 Hz/50 A | 4.0480 | −0.0020 |
| 10 Hz/100 A | 4.0447 | −0.0054 |
| 10 Hz/150 A | 4.0452 | −0.0073 |

*3.4. Effect of Low-Frequency Electromagnetic Field on Micro-Structure and Precipitate of Aged Alloys*

A typical micro-structure of 2A97 alloy after T6 treatment cast in conventional DC casting process is shown in Figure 9a, which consists of elongated, strip-like grains (22.5 ± 12.2 μm in width) aligned in the direction of extrusion, and coarse second-phase particles distributed along the grain boundaries. The average width of strip-like grains in the alloy cast under an electromagnetic field with 10 Hz/150 A decreases to 18.1 ± 8.9 μm as shown in Figure 9b. A large number of fine second-phases are found within the strips. Based on the previous discussion in Section 3.1, it is evident that grain size and morphology are strongly affected by the presence of an electromagnetic field, which is obviously hereditary from the casting, extruding, and aging stages.

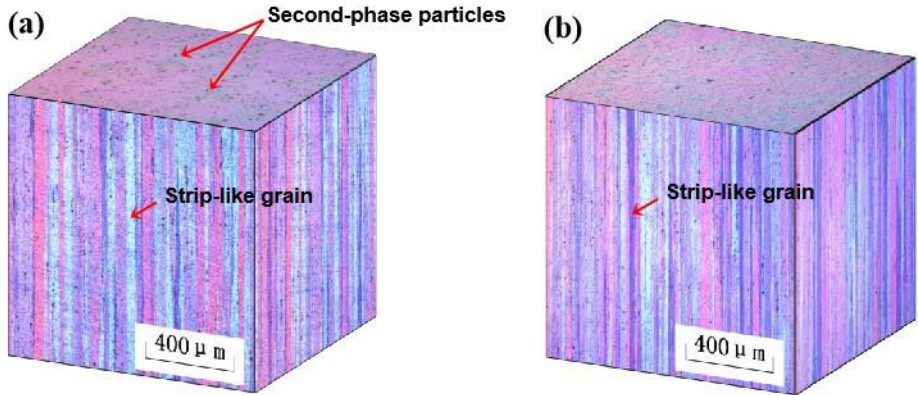

**Figure 9.** Three-dimensional micrographs of 2A90 alloy after T6 treatment. (**a**) Conventional DC casting; (**b**) LFEC with 10 Hz/150 A.

The precipitates which played a crucial role in strengthening the Al-Cu-Li alloy were identified to be $T_1$ ($Al_2CuLi$), $\theta'$ ($Al_2Cu$), $S'$ ($Al_2CuMg$), $\beta'$ ($Al_3Zr$), particularly $T_1$ phases [18,19]. In this study, the morphology of $T_1$ phases in the 2A97 aged alloy were characterized by TEM as shown in Figure 10. The density and length of $T_1$ phases were measured by using Image-Pro Plus software, the results are shown in Figure 11. The lengths of the plate-shape precipitates have been categorized into various groups. In the aged alloy prepare by LFEC with 10 Hz/150 A, the density of $T_1$ phases less than 150 nm increased in each size group. It is obvious seen that the length of $T_1$ phases between 100 nm to 150 nm is dominant, and the amount of $T_1$ phases whose length exceeding 250 nm is small. After applying electromagnetic field with 10 Hz/150 A, the total density of $T_1$ phase increased from 48.2 plates/$\mu m^2$ to 63.1 plates/$\mu m^2$, and the average length of $T_1$ phase was decreased from 162.3 nm to 120.5 nm. In general, the statistical results show that under the influence of a specific low frequency electromagnetic field, $T_1$ phases became finer, exhibiting a higher density and more uniform distribution.

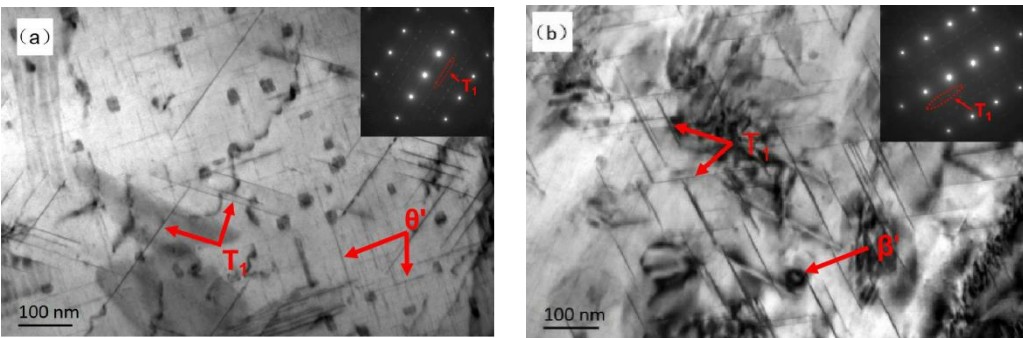

**Figure 10.** TEM bright field micrographs and diffraction patterns of 2A90 alloy after T6 treatment taken along $\langle 112 \rangle_{Al}$ direction (**a**) Conventional DC casting; (**b**) LFEC with 10 Hz/150 A.

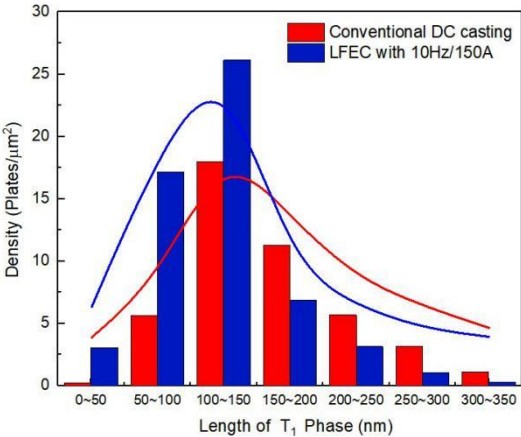

**Figure 11.** Number density of $T_1$ of 2A97 alloy after T6 treatment.

*3.5. Effect of Low-Frequency Electromagnetic Field on Mechanical Properties of Aged Alloys*

Figure 12a,b illustrate the relationship of the tensile properties of 2A97 alloy after T6 treatment on the current intensity of the magnetic field. It is clear that both the ultimate tensile strength (UTS) and yield strength (YS) of the alloys were improved significantly under the influence of an electromagnetic field. With the electromagnetic field of 10 Hz/150 A, the ultimate tensile strength of the alloy reached its highest value, 595.5 MPa, which is 53.2 MPa higher than that without magnetic field. In all conditions the elongation of the alloy kept higher 8.5%. The reasons for the low tensile strength of the 2A97 aged alloy prepare by conventional DC casting are mainly attributed to the presence of casting defects and serious segregation along grain boundaries. The coarse non-equilibrium eutectic phases in grain boundaries easily form coarse precipitates during the aging treatment due to the high content of alloy elements. These coarse precipitates will break away from the matrix and form micro-cracks under the action of tensile stress. However, the 2A97 alloy cast by the LFEC method obtains better mechanical properties after T6 treatment due to grain refinement and precipitation strengthening.

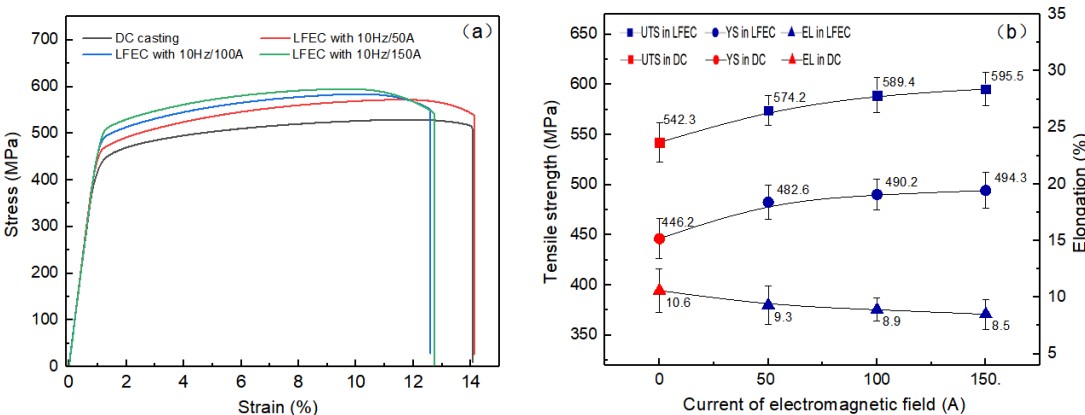

**Figure 12.** Engineering stress-strain curves and tensile properties of 2A97 aged alloy cast in different electromagnetic field conditions.

## 4. Discussion

### *4.1. The Mechanism of Effect of LFEC on the Micro-Structure*

The mechanism of grain refinement under low-frequency electromagnetic field has been systematically and extensively investigated [9–12]. In this treatment, the melt in the sump was

subjected to Lorenz forces caused by the interaction of the induced current and magnetic field [20]. The Lorentz force expressed as follows:

$$f = \frac{1}{\mu}(B \cdot \nabla)B - \frac{1}{2\mu}\nabla B^2 \tag{3}$$

where $B$ and $J$ are the magnetic induction intensity and current density generated in the melt, $\mu$ is the permeability of the melt. The first term on the right hand of Equation (3) is a rotational component which results in a forced convection and flow in the melt. The second term is a potential force balanced by static pressure of the melt.

The vigorous vortex agitation produced by the forces not only accelerate the melt exchange between the internal and the external of the sump, but also intensify the heat transfer between the melt and graphite ring [21], which will reduce the temperature gradient and shallow liquid sump depth as shown in Figure 13. The forced convection makes the low temperature melt near the mold move into the center, and high temperature melt into border, which makes the temperature fields in the melt become more uniform [22]. The fast removal of heat along the solidification front enhances the tendency of the liquid to undercool, which results in activating of a large number of nucleation for crystallization. Fast flow of melt along the solidification front suppresses the growth of dendrite. Moreover, the dendrite fragments are detached from the solidification front and carried away into the sump with fast flow of melt forming nuclei for crystallization [23]. The nucleation takes place almost simultaneously in the mold and the nuclei bump, and frictionate each other which is helpful for the formation of equiaxed grains [22]. As a result, the micro-structure of the as-cast ingots prepared by LFEC are fine equiaxed and more uniform than that of a conventional DC ingot.

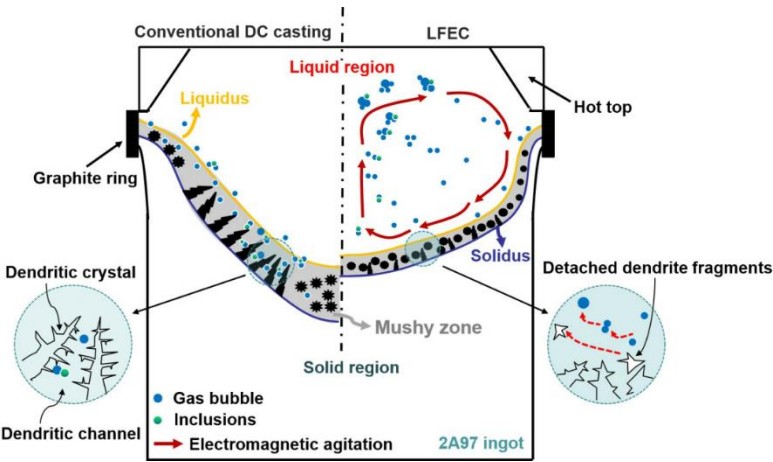

**Figure 13.** Schematic diagrams of the solidification.

When lithium is added into molten aluminum alloy, the molten alloy reacts with $H_2O$ in the atmosphere and oxidizes quickly, then absorbs the hydrogen decomposed from $H_2O$ [4,7]. If hydrogen and other gases fail to achieve a timely expulsion, they will be trapped into dendritic channel during solidification. Furthermore, the liquid Al-Li alloys are easily subjected to the oxygenation on the surface under the atmospheric conditions and unfavorable protection, which results in forming a symbiosis of slag inclusion and gas bubble. These slag inclusions and gas bubbles remained in the ingot forming casting defects will deteriorate the mechanical properties of the formed alloys. However, when an electromagnetic field was applied, the strong Lorenz forces produce vigorous stirring in the melt during the cast process. Then, gas bubbles collide with each other, coalesce to grow, and get away from the solidification front with the fast flow of melt [24].

### 4.2. The Mechanism of Effect of LFEC on the Solid Solubility

During non-equilibrium solidification, the solidified grains continuously discharge alloying elements towards liquid through the solid-liquid interfaces. With the solidification process, the liquid containing a high concentration of alloying elements will remain inter-dendritic when dendrites contact. Finally, a large amount of alloying elements accumulate along the grain boundaries and form non-equilibrium eutectic phase with a low melting point, namely grain boundary segregation. The serious grain boundary segregation will easily form various coarse precipitates. The other form of alloying elements is a solid solution in the matrix and precipitation within grains [25]. When electromagnetic field was applied, the heat transfer of the melt in the mold was intensified, resulting in deep under-cooling and fast solidification. An accelerated solidification rate means that solute elements have no sufficient time to be discharged from the solid phase to the liquid phase, hence a large number of alloying elements are retained within grains. According to the equation of the effective partition coefficient $k_e$ derived by Boton, Prim, and Slichter [26]:

$$k_e = \frac{\rho_S}{\rho_L} = \frac{k_0}{k_0 + (1 - k_0)e^{-R\delta/D}} \tag{4}$$

where $\rho_S$ is the concentration of solute elements in the solid phase, $\rho_L$ is the concentration of solute elements in the liquid phase, $k_0$ is the solute equilibrium partition coefficient, $R$ is the solid/liquid interface movement velocity, $\delta$ is the boundary layer thickness, and $D$ is the diffusion coefficient. The improvement in the cooling intensity increases the solid/liquid interface movement velocity. Meanwhile, under the electromagnetic field the charged particles such as $Al^{3+}$, $Cu^{2+}$, $Li^+$, $Mg^{2+}$, and $Mn^{2+}$ in the liquid phase will be subjected to different Lorentz forces because these charged particles have different charges and atomic masses, which can lead to irregular movement between the charged particles [27]. This irregular movement of solute atoms in the boundary layer will restrain the diffusion of solute atoms along the direction perpendicular to the solid-liquid interface, namely increasing the boundary layer thickness $\delta$ as shown in Figure 14.

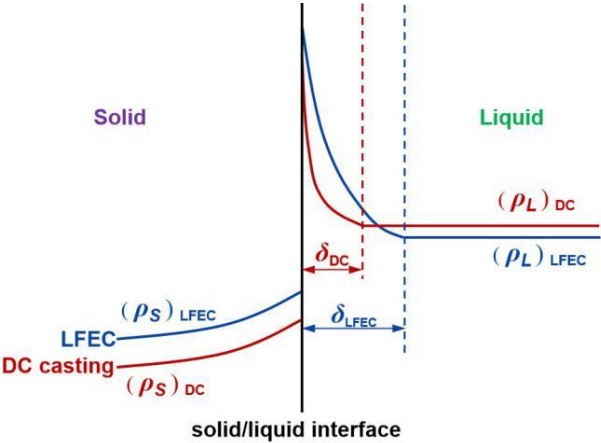

**Figure 14.** Distribution of alloying element in area of solid/liquid interface.

With respect to energy, it is believed that an electromagnetic field can provide greater energy to meet the requirements of crystal lattice distortion owing to the solid solution of more alloying elements in a matrix. Consequently, the LFEC can increase the effective partition coefficient $k_e$ and the solid solubility within grains.

Based on the above discussion, it is evident that the solid solubility of alloying elements are strongly affected by the presence of an electromagnetic field. For 2A97 alloy, an increase in the number of intra-granular Mg and Mn atoms reduces the stacking fault energy of the alloys and proves conducive to the formation of stacking faults [28,29]. This also provides favorable conditions for the formation of

G.P zone (solute atomic agglomeration zone) and the nucleation of the precipitation phase. Moreover, an increase in the number of Cu and Li atoms within grains enhances super-saturation in the matrix, which provides the necessary conditions for the precipitation of the strengthening phase. Further, with an increase in the amount of Mg, a larger number of super-saturated vacancies were created which gave rise to the formation of Mg-V (vacancies). In this scenario, Li atoms could not compete for Mg-V (vacancies), and could also not pass through them. Hence, Li diffusion is prevented and the growth of $T_1$ (Al$_2$CuLi) phase is suppressed [29,30]. This mechanism of promoting the strength of nucleation and inhibiting $T_1$ phase growth is an important factor for obtaining a large number of fine, uniformly-distributed $T_1$ phases in low frequency electromagnetic casting.

## 5. Conclusions

In this study, the effect of a low-frequency electromagnetic field on the micro-structure and tensile properties of 2A97 Al-Li alloys was studied by comparing the process of conventional DC casting and low-frequency electromagnetic casting. The results can be summarized as follows:

(1). The micro-structure of ingots was remarkably refined by using LFEC. The average grain size of a one-half radius region of a cross section of ingots underwent refinement from 303.2 μm in conventional DC casting to 119.6 μm in LFEC with 150 A/10 Hz, and the grain morphology changed from coarse and rose dendrites to uniform and fine equiaxed structures.

(2). The number of casting defects decreased remarkably in the as-cast ingot prepare by LFEC.

(3). The solid solubility of alloying elements within grains in the as-cast ingot prepared by LFEC was significantly enhanced.

(4). The $T_1$ phases in the aged alloy prepared by LFEC became finer and exhibited a higher density and more uniform distribution.

(5). After applying a low frequency electromagnetic field during DC casting, the mechanical properties of 2A97 aged alloys were improved significantly. The maximum ultimate tensile strength, 595.58 MPa, the yield strength of 494.31 MPa, and the elongation of 8.47% was obtained when the parameters of LFEC were 10 Hz and 150 A.

**Author Contributions:** Conceptualization, J.C. and X.W.; methodology, J.C.; validation, J.C.; formal analysis, X.W. and F.W.; investigation, F.W.; resources, X.W. and J.C.; data curation, F.W.; writing—original draft preparation, F.W.; writing—review and editing, F.W. and X.W.; visualization, F.W.; supervision, X.W.; project administration, J.C. and X.W.; funding acquisition, J.C. and X.W.

**Funding:** This research was funded by National Key Research and Development Program of China (grant number 2016YFB0300901), and the National Natural Science Foundation of China (grant number U1708251, 51574075, U1608252), the Fundamental Research Funds for the Central Universities (grant number N180905010).

**Conflicts of Interest:** The authors declare no conflict of interest.

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
