# Peer review of "Micro-Structure and Mechanical Properties of 2A97 Al-Li Alloy Cast by Low-Frequency Electromagnetic Casting"

_metals, doi:10.3390/met9080822_

Round 1
Reviewer 1 Report
The problem presented by the authors is interesting for the wider audience. Many institutions around the globe are working on a similar problem, but no plagiarism has been detected, beside that the majority of them are cited by the authors.The paper has some flaws. The main problem is with editing, the authors have not worked on the technical quality of the paper enough. There are some moderate mistakes like e.q.
page 1 paragraph 29- ...because of the molten- should be -because the molten....
The paper has different fonts all along the whole paper like:
page 6 paragraph 143 to 152
page7 paragraph 180
page 9 paragraph 213
page11 paragraph 250
page 11 paragraph 260 to 261
page 12/13 paragraph 305 to 310
and many more
The formulas used in the paper have different font sizes and number sizes also.
The references have numbers without and with brackets for the second time.
The quality of the majority of figures should be improved like e.q. figure 5 and 7
Author Response
Dear reviewer:
All the authors of this manuscript are grateful to you and the Reviewer for the valuable comments and suggestions. We have made revisions with blue font and underline in the marked-up manuscript, according to the comments and suggestions from the Reviewers. The primary revisions and the responds to the reviewer’s comments are as flowing:
Responds to the reviewer’s comments and suggestions:
1.The comment ( The authors have not worked on the technical quality of the paper enough. There are some moderate mistakes like e.q. …)
Response: Thanks a lot for the well-read and careful reviewer. Page1 paragraph 29: " because of the molten..." has been replaced by " because the molten...".
2. The comment ( The paper has different fonts all along the whole paper like: …)
Response: We are sorry for the our poor mistake. We have adjusted the fonts all along the whole paper.
3. The comment ( The formulas used in the paper have different font sizes and number sizes also)
Response: We are sorry for the mistake in the formulas. We have adjusted the fonts in formulas.
4. The comment ( The references have numbers without and with brackets for the second time)
Response: Thanks a lot for the careful reviewer. We have deleted the numbers with brackets in the references section.
5. The comment ( The quality of the majority of figures should be improved like e.q. figure 5 and 7)
Response: It is quite right as the reviewer commented. We have adjusted the bright and contrast of figure 5 and 7 for the reader can see them clearly.
We appreciate for reviewer’s warm work earnestly, and hope that the correction will meet with approval. We will never give up the chance to improve the quality of our manuscript.
Sincerely
Fuyue Wang
Reviewer 2 Report
The manuscript titled Micro-structure and mechanical properties of 2A97 3 Al-Li alloy cast by low-frequency electromagnetic 4 casting investigates the material characteristics of Al- Li alloy from mechanical point of view. The manuscript is well written and easy to follow. Before the acceptance of paper the following changes are recommended. It is hard to see the scale in Fig 6 and Fig 7. Line 347 “is follows” Line 229 micrographs Line 324 as shown Line 349 unconventional.
Author Response
Dear reviewer:
All the authors of this manuscript are grateful to you and the Reviewer for the valuable comments and suggestions. We have made revisions with blue font and underline in the marked-up manuscript, according to the comments and suggestions from the Reviewers. The primary revisions and the responds to the reviewer’s comments are as flowing:
1.The comment ( It is hard to see the scale in Fig 6 and Fig 7.)
Response: We are sorry for the small scale in figure 6 and figure 7. We have replaced the scale in every picture.
2.The comment ( Line 347 “is follows”, Line 229 micrographs, Line 324 as shown, Line 349 unconventional.)
Response: Thanks a lot for the well-read and careful reviewer. We have tried to avoid the paper’s language defects for the readability in the revised form.
Line 347: " The summary of the results is as follow " has been replaced by " The summary of the results is follows"
Line 229: The misspelled word has been corrected.
Line 324: "... as show in Fig. 5" has been replaced by "... as shown in Fig. 5"
Line 349: " The average grain size of 1/2 radius region of ingots underwent refinement from 303.2 µm inconventional DC... " has been replaced by " The average grain size of 1/2 radius region of ingots underwent refinement from 303.2 µm in conventional DC..."
We appreciate for reviewers’ warm work earnestly, and hope that the correction will meet with approval. We will never give up the chance to improve the quality of our manuscript.
Sincerely
Fuyue Wang
Reviewer 3 Report
A separate file with comments and suggestions was uploaded

Round 2
Reviewer 3 Report
Dear Authors
Although I don't consider your anwers and modifications to the manuscript particularly good, I beleive the article has now quality to be published. Nevertheless, editing your article would be important to correct numerous grammar and spelling errors in the English writting.